# PruneHal: Reducing Hallucinations in Multi-modal Large Language Models through Adaptive KV Cache Pruning

## Abstract

While multi-modal large language models (MLLMs) have made significant progress in recent years, the issue of hallucinations remains a major challenge. To mitigate this phenomenon, existing solutions either introduce additional data for further training or incorporate external or internal information during inference. However, these approaches inevitably introduce extra computational costs. In this paper, we observe that hallucinations in MLLMs are strongly associated with insufficient attention allocated to visual tokens. In particular, the presence of redundant visual tokens disperses the model's attention, preventing it from focusing on the most informative ones. As a result, critical visual cues are often under-attended, which in turn exacerbates the occurrence of hallucinations. Building on this observation, we propose **PruneHal**, a training-free, simple yet effective method that leverages adaptive KV cache pruning to enhance the model's focus on critical visual information, thereby mitigating hallucinations. To the best of our knowledge, we are the first to apply token pruning for hallucination mitigation in MLLMs. Notably, our method don't require additional training and incurs nearly no extra inference cost. Moreover, PruneHal is model-agnostic and can be seamlessly integrated with different decoding strategies, including those specifically designed for hallucination mitigation. We evaluate PruneHal on several widely used hallucination evaluation benchmarks using four mainstream MLLMs, achieving robust and outstanding results that highlight the effectiveness and superiority of our method. Our code will be publicly available.

## 1 Introduction

Recent advancements in multi-modal large language models (MLLMs) have led to significant breakthroughs, enabling these models to effectively tackle a wide range of complex visual tasks (Wang et al., 2024d; Bai et al., 2023; Achiam et al., 2023; Chen et al., 2024b; Dai et al., 2023; Liu et al., 2023b; 2024a; Lu et al., 2024; Yao et al., 2024). Despite MLLMs' strong performance across various visual tasks, their practicality remains limited due to hallucinations. The issues of hallucinations (Bai et al., 2024; Rohrbach et al., 2018; Li et al., 2023b) often cause MLLMs to output content that is inconsistent with visual input, severely undermining their reliability.

Several studies have explored mitigating hallucinations in MLLMs from different perspectives. Some works (Gunjal et al., 2024; Zhou et al., 2023; Liu et al., 2023a; Yu et al., 2024) focus on further training on specifically designed datasets or alignment to reduce the model's propensity to generate hallucinated content, typically by leveraging additional annotated data or specialized supervision. In contrast, training-free approaches (Leng et al., 2024; Huang et al., 2024; Liu et al., 2024b; Wang et al., 2024c; Xu et al., 2025) aim to alleviate hallucinations during inference, often by designing specific decoding strategies. Although these methods have demonstrated effectiveness, they incur additional training costs or computational overhead during inference.

In this paper, we first note that **hallucinations in MLLMs are closely associated with the model's insufficient attention to visual tokens.** Previous works (Chen et al., 2024a; Liu et al., 2024b) have demonstrated that in MLLMs, visual tokens constitute most of the input tokens, yet they receive little attention during the forward pass of the self-attention (Vaswani et al., 2017). It is reasonable

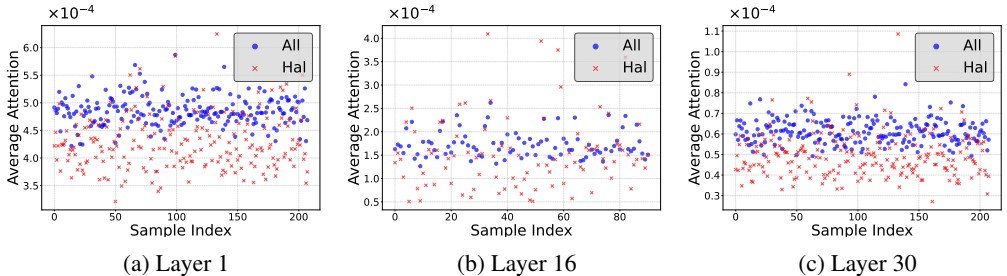

Figure 1: An example where the hallucinated word "bed", which is not presented in the image, is correctly changed to "window", by amplifying the visual attention. More examples can be found in Appendix. C.3.

to hypothesize that the generation of hallucinated content may also be associated with this phenomenon. To intuitively validate this, we conduct a qualitative experiment. Specifically, we attempt to directly amplify the visual attentions in attention maps during model inference, immediately before the model generates hallucinated content (details are provided in Appendix. C.1). As shown in Fig. 1, increasing MLLMs' attention to visual information enables it to generate content that aligns with the image during inference process, thereby reducing hallucinations.

To illustrate this quantitatively, we present a comparison between the visual attention in the inference steps that lead to hallucinations and the average visual attention across all inference steps (details are provided in Appendix. C.2). As illustrated in Fig. 2, it is clear that across the shallow, middle, and deep layers of LLaVA-v1.5-7B, the generation of hallucinated contents are strongly associated with lower attention scores on visual tokens (intuitively, most of the blue scatter points lie above the red ones). These evidences reveal that the issue of lower attention scores received by visual tokens is potentially one of the cause of generating hullucinated contents during decoding.

|  (a) Layer 1 | (b) Layer 16 | (c) Layer 30 |
|---|---|---|

Figure 2: Average visual attention in language model of LLaVA-v1.5-7B. Each blue scatter point represents the mean attention scores received by all visual tokens in a caption, averaged over all decoding steps. The red scatter points represent the mean attention values when hallucinated content is generated in captions. Attention scores are averaged over all attention heads. Images and captions are selected from the validation set of MSCOCO 2014.

Meanwhile, existing works (Chen et al., 2024a; Yang et al., 2025) have pointed out that visual tokens in MLLMs exhibit widespread redundancy, with only a small number of key tokens containing critical visual information needed for inference. We reasonably hypothesize that **hallucinations in MLLMs are highly correlated with visual tokens' redundancy**. In particular, the redundancy of visual tokens could disperse models' attention and cause important tokens to be under-attended. This further reduces attention to critical and informative visual tokens, exacerbating the hallucination phenomenon.

Based on these insights, we propose **PruneHal**, a novel framework that leverages KV cache pruning to mitigate hallucinations in MLLMs. Initially, we apply a simple KV cache pruning method that retains the top-K most important visual tokens. We show that this top-K-based approach is both simple and effective, enabling the model to focus on key visual tokens. However, this strategy

struggles to determine the optimal time for pruning, often leading to excessive pruning. This, in turn, results in the loss of crucial visual information, negatively impacting the overall performance of the model. To address this, we propose an adaptive pruning strategy that tracks the historical visual attention distribution to prune redundant visual tokens more effectively. This allows PruneHal to achieve a balanced trade-off as inference progresses, mitigating hallucinations while preserving output quality.

To verify the effectiveness of the proposed method, we conducted extensive experiments on multiple widely-used MLLMs, including LLaVA (Liu et al., 2023b), InstructBLIP (Dai et al., 2023), and Qwen-VL (Bai et al., 2023), across various benchmarks and hallucination metrics. The results demonstrated that PruneHal significantly improves performance on a range of MLLMs, benchmarks and datasets, showing favourable robustness. Additionally, we found that PruneHal is fully compatible with existing decoding strategies designed specifically for hallucination mitigation.

Our contribution can be summarized as follows:

1. Our experiments reveal a strong link between hallucinations in MLLMs and the insufficient attention given to key visual tokens. We further observe that redundant visual tokens siphon off a substantial portion of the model's attention, leaving key visual tokens under-attended.

2. To address the above issues, we introduce adaptive KV cache pruning to mitigate this issue, and propose PruneHal framework. Our PruneHal alleviates hallucinations in MLLMs during inference time while introducing almost no computational overhead.

3. Experimental results show that our PruneHal effectively mitigates MLLMs' hallucinations. When combined with existing approaches, PruneHal achieved state-of-the-art results on various hallucination benchmarks.

## 2 RELATED WORKS

### 2.1 HALLUCINATION IN MULTI-MODAL LARGE LANGUAGE MODELS

Hallucinations in MLLMs refer to the phenomenon where the visual content in the input conflicts with the generated textual information. Existing works have attempted to detect and address hallucinations in MLLMs from multiple perspectives. CHAIR (Rohrbach et al., 2018) requires large models to generate detailed captions for images and calculates the proportion of hallucinated content in the captions. POPE (Li et al., 2023b) turns hallucination problem into binary classification which can detect object hallucinations in MLLMs.

Existing works mainly address hallucinations in MLLMs from two perspectives. On one hand, some approaches involve additional training or use external knowledge to guide the model. LURE (Zhou et al., 2023) trains an extra state detector that triggers the regeneration of detected hallucinated contents by a revisor model; WoodPecker (Yin et al., 2024) introduces an additional visual model to monitor and ask the original model to regenerate hallucinated contents. On the other hand, some works focus on specially designed decoding strategies: OPERA (Huang et al., 2024) highlights the relationship between MLLM's aggregation patterns and hallucinations, and mitigates hallucinations using over-trust logit penalty and retrospection; VCD (Leng et al., 2024) points out that models' visual uncertainty may lead to hallucinations and proposes a contrastive decoding method to address this; DeCo (Wang et al., 2024c) suggests that MLLMs can correctly perceive visual content, and leverages information from shallow layers of language models in MLLMs to guide the decoding process. However, all above mentioned works introduce additional computational overhead, thus slows down model inference speed.

### 2.2 VISUAL TOKEN COMPRESSION IN MULTI-MODAL LARGE LANGUAGE MODELS

Previous works accelerate MLLM's inference by compressing visual tokens. FastV (Chen et al., 2024a) firstly leveraged attention scores in language models of MLLMs to prune redundant visual tokens. LLaVA-PruMerge (Shang et al., 2024) and VTC-CLS (Wang et al., 2024b) utilize information from the `[CLS]` token in MLLM's visual encoder to prune and merge redundant visual tokens. VisionZip (Yang et al., 2025) picks a small proportion of key visual tokens based on attention scores extracted from the visual encoder and applies a merging strategy to retain the remaining information.

Additionally, some works (Tao et al., 2025; Fu et al., 2024; Sun et al., 2025) have been specifically designed for video LLMs. These methods effectively accelerate MLLM's inference.

## 3 METHODOLOGY

### 3.1 PRELIMINARIES

During MLLMs' inference, for each token generation, the language model performs a forward pass. When KV cache is available, the self-attention modules computes the attention map using only the last generated token as the query. During decoding process, we denote the input sequence as $\mathbf{X} = [\mathbf{x}_1, \mathbf{x}_2, ..., \mathbf{x}_n] \in \mathbb{R}^{n \times d}$, where $\mathbf{x}_n \in \mathbb{R}^{1 \times d}$ is the last generated token. $d$ denotes the dimension of hidden states.

In self-attention modules, for the $i$-th attention head, the attention map $\mathbf{A}^i \in \mathbb{R}^{1 \times n}$ for the last generated query token is computed as follows:

$$\mathbf{A}^i = \texttt{softmax}\left(\frac{\mathbf{q}_n^i {\mathbf{K}_n^i}^T}{\sqrt{d_k}}\right), \tag{1}$$

where $\mathbf{q}_n^i \in \mathbb{R}^{1 \times d_k}$ is the query vector for the last generated token, $\mathbf{K}_n^i$ is the key matrix of the $i$-th attention head, and $d_k$ denotes the dimension for each attention head. Next, the key and value vectors $\mathbf{k}_n^i$ and $\mathbf{v}_n^i$ are concatenated with the corresponding KV cache and the cache itself is updated:

$$\mathbf{K}_{cache}^i := \texttt{concat}([\mathbf{K}_{cache}^i, \mathbf{k}_n^i]), \quad \mathbf{V}_{cache}^i := \texttt{concat}([\mathbf{V}_{cache}^i, \mathbf{v}_n^i]), \tag{2}$$

where $\mathbf{K}_{cache}^i, \mathbf{V}_{cache}^i \in \mathbb{R}^{n \times d_k}$ represent the KV cache.

We average the attention map $\mathbf{A}^i$ over all attention heads to obtain the average attention map $\mathbf{A} \in \mathbb{R}^{1 \times n}$, where each value in $\mathbf{A}$ represents the attention paid by the last generated token to the token at that position. Attention map $\mathbf{A}$ can also be represented as:

$$\mathbf{A} = \{\mathbf{A}_t, \mathbf{A}_v, \mathbf{A}_o\}, \tag{3}$$

where $\mathbf{A}_t, \mathbf{A}_v, \mathbf{A}_o$ denotes attention scores distributed to prompt text tokens, visual tokens and output text tokens, respectively. Previous works (Chen et al., 2024a; Liu et al., 2024b) have highlighted that attention scores in $\mathbf{A}_v$ are often very low, especially when considering that the number of visual tokens is often large.

In Sec. 1, we have explored the connection between this phenomenon and hallucinations in MLLMs. In this section, **we propose PruneHal, a training-free, plug-and-play framework to mitigate MLLMs' hallucination**. Specifically, in Sec. 3.2, we propose to leverage simple top-K-based KV cache pruning to guide MLLMs toward focusing on critical visual tokens. In Sec. 3.3, we introduce adaptive design into our framework, which seeks to strike a balance between loss of crucial visual information and eliminating the redundant visual tokens.

### 3.2 KV CACHE PRUNING REMOVES REDUNDANT VISUAL TOKENS TO ENHANCE FOCUS ON CRITICAL VISUAL INFORMATION

In this subsection, we argue that the insufficient attention paid to critical visual tokens is a key contributor to hallucinations in MLLMs. Specifically, a large number of redundant visual tokens consume a substantial portion of the model's attention, leaving key visual information under-attended.

Simply amplifying visual attention as in Fig. 1 not only leads MLLMs to generate uninformative outputs, but also exacerbates the disruption caused by redundant tokens. To address this, we naturally turn to KV cache pruning, which preserves the model's language prior while seamlessly removing redundant visual tokens. Among all visual tokens with attention scores $\mathbf{A}_v$, we select the top-$k$ tokens with the highest attention scores:

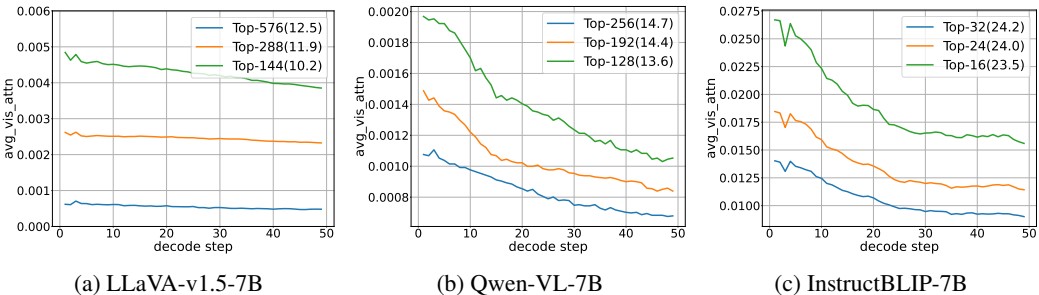

(a) LLaVA-v1.5-7B  (b) Qwen-VL-7B  (c) InstructBLIP-7B

Figure 3: Average visual attention scores over all samples from the validation set of MSCOCO 2014. The x-axis represents decoding steps, and the y-axis represents the average visual attention score. We show data from Layer 1 for all three models. In the figures, **Top-$K$ denotes retaining $K$ most attended visual tokens after pruning, corresponding to the curve of that color**. The curves start from the second decoding step after pruning, and ends at 50-th decoding step. The numbers in parentheses represent the proportion of hallucinated objects among all mentioned objects; the higher the numbers, the more severe the hallucinations. More examples can be found in Appendix. C.4.

$$\mathcal{I}_v = \mathtt{TopK}(\mathbf{A}_v, \ k), \ 0 < k < N_v, \tag{4}$$

where $N_v$ denotes the number of visual tokens. After obtaining the indices of the tokens to be retained in the previous decoding step, we perform KV cache pruning in the current step by keeping only the entries corresponding to those indices. For convenience, we only show the case of a single attention head; the principle of multi-head attention is exactly the same:

$$\mathbf{K}'_{cache} = \{\mathbf{K}_{text}, \mathbf{K}'_{vis}\}, \ \text{where} \ \mathbf{K}'_{vis} = \mathbf{K}_{vis}[\mathcal{I}_v, :] \tag{5}$$

$$\mathbf{V}'_{cache} = \{\mathbf{V}_{text}, \mathbf{V}'_{vis}\}, \ \text{where} \ \mathbf{V}'_{vis} = \mathbf{V}_{vis}[\mathcal{I}_v, :], \tag{6}$$

where we can represent $\mathbf{K}_{cache} = \{\mathbf{K}_{text}, \mathbf{K}_{vis}\}$, $\mathbf{K}_{text}$ and $\mathbf{K}_{vis}$ correspond to the KV cache for text and visual tokens, respectively. Here $\mathbf{K}_{vis}, \mathbf{V}_{vis} \in \mathbb{R}^{N_v \times d}$ represent full KV cache, while $\mathbf{K}'_{vis}, \mathbf{V}'_{vis} \in \mathbb{R}^{k \times d}$ represent pruned KV cache. Among all KV cache entries in storage, we retain the subset corresponding to the indices in $\mathcal{I}_v$ and discard the others.

As stated above, we conducted simple top-K-based KV cache pruning[1] on several different MLLMs. During the first decoding process, we recorded the indices to keep in the first decoding process, and conduct pruning in the second decoding process. To validate the effectiveness of this, during each subsequent decoding step, we record the visual attention distribution under different degrees of KV cache pruning, averaged over the 100 random selected samples from the validation set of MSCOCO 2014. The visual attention plots are shown below in Fig. 3.

As shown in Fig. 3, KV cache pruning removes redundant visual tokens, which increases the average attention scores of the remaining ones. Moreover, as more tokens are discarded and visual attention scores increase, hallucinations are gradually mitigated, demonstrating that pruning effectively guides MLLMs to focus on critical visual cues. Such an effect is consistent across all layers and models.

### 3.3 ADAPTIVE KV CACHE PRUNING STRIKES A TRADE-OFF BETWEEN PRESERVING VISUAL INFORMATION AND MITIGATING HALLUCINATIONS

While KV cache pruning can mitigate hallucinations, excessive pruning leads to substantial loss of visual information, which in turn inevitably degrades the performance of MLLMs (Wang et al., 2024a; Liu et al., 2024c; Zhang et al., 2023). Moreover, as shown in Fig. 3, **while the number of tokens in the auto-regressive sequence keeps growing as decoding step increases, attention scores assigned to visual tokens gradually diminishes, causing the model's focus on them to**

---

[1]We want to showcase conceptual effectiveness when combining KV Cache pruning to mitigate MLLMs' hallucinations. More advanced KV Cache pruning methods are also compatible in our framework, which we leave as our future works.

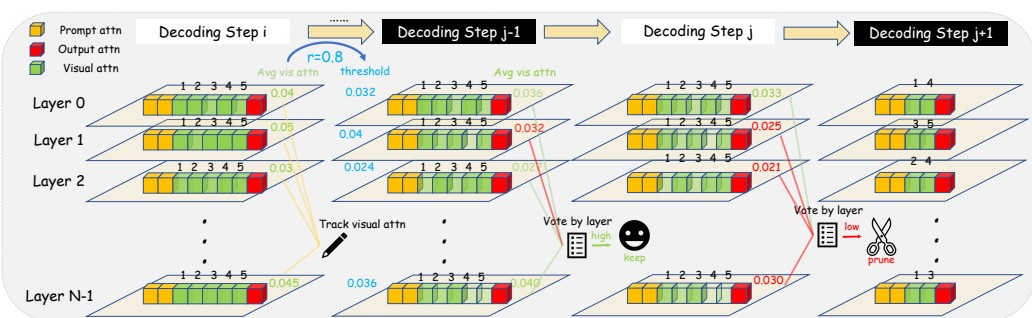

Figure 4: Illustration of PruneHal. During decoding step *i*, visual attention distribution are tracked and continuously monitored in subsequent steps. Once the visual attention distribution of more than half of the layers falls below a predefined threshold, a pruning operation is performed.

**continuously decline.** This highlights the need for a dynamic mechanism to sustain attention to visual tokens throughout decoding.

Therefore, we emphasize the need to dynamically strike a balance during decoding between preserving crucial visual information and eliminating redundant visual tokens to maintain the model's performance. To address this, our PruneHal framework adaptively performs KV cache pruning, enhancing visual attention while avoid crucial visual information loss caused by excessive pruning, thereby mitigating hallucinations while retaining MLLMs' performance to the maximum extent.

To this end, we propose to adaptively conduct KV cache pruning. As illustrated in Fig. 4, we adopt voting mechanism to decide whether to prune. Specifically, when a majority of layers indicate insufficient visual attention, a pruning operation is triggered to encourage the model to concentrate on the most informative visual tokens. Each pruning step removes a fixed proportion of the remaining tokens, and an upper bound is imposed on the total number of pruning operations to prevent excessive loss of visual information.

At the first decoding step, we record the average attention score of all remaining visual tokens in each layer, i.e., $\mathrm{avg}(\mathbf{A}_{v_i}^1)$ for $i = 1, 2, \ldots, N$, where $N$ is the number of layers in the language model. In the second step, these recorded values are used to guide the first pruning operation, as stated in Eq. 4–6.

---

**Algorithm 1:** Our PruneHal framework

**Input:** keep ratio $r$; max pruning times $t$; KV cache $\mathbf{K}_{vis}, \mathbf{V}_{vis} \in \mathbb{R}^{N_v \times d}$; visual token count $n$.

**Input:** $\{\mathbf{A}_{v_i}^m\}_{i=1}^N, m = 1, 2, \ldots$; $i$: layer index, $m$: decoding step

1  $prune\_cnt \leftarrow 0, \{\mathcal{A}_i\}_{i=1}^N \leftarrow \{\mathbf{A}_{v_i}^1\}_{i=1}^N$;

2  **for** *each decoding step $m$* **do**

3     **if** *prune_cnt = t* **then**

4        **continue**;

5     $\mathcal{I}_l = \{i \mid \mathbf{A}_{v_i}^m < \sqrt{r} \cdot \mathcal{A}_i\}$; // `layer vote`

6     **if** $|\mathcal{I}_l| \geq \frac{N}{2}$ *or $m = 2$* **then**

7        **for** $i = 1$ **to** $N$ **do**

8           $\mathcal{I}_v = \mathrm{TopK}(\mathbf{A}_{v_i}^m, r \times n) \subset \{1, 2, \ldots, n\}$;

9           $\mathbf{K}_{vis} \leftarrow \mathbf{K}_{vis}[\mathcal{I}_v, :]$;

10          $\mathbf{V}_{vis} \leftarrow \mathbf{V}_{vis}[\mathcal{I}_v, :]$; // `Perform KV cache pruning`

11          $\mathcal{A}_i \leftarrow \mathbf{A}_{v_i}^{m-1}$; // `Update parameters`

12       $prune\_cnt \leftarrow prune\_cnt + 1$; // `Update prune count`

13       $n \leftarrow r \times n$; // `Update visual token count`

---

Our full framework can be described in Alg. 1. After each pruning operation, the historical visual attention distribution are immediately refreshed to reflect the most recent state. We predefine the retention ratio for KV cache pruning as $r$, which controls the fraction of visual tokens to be preserved

at each step. In subsequent decoding steps, the framework continuously monitors average visual attention scores across layers. Whenever more than half of the layers show that their current average visual attention scores have dropped below $\sqrt{r}$ times the historical distribution, a pruning operation is triggered. During this process, only the top-$r$ fraction of the remaining visual tokens with the highest attention scores are retained in the KV cache, while the remaining tokens are discarded.

By performing pruning dynamically, we minimize the loss of crucial visual information while maintaining the model's focus on key visual tokens. In this way, the model's all-round ability is preserved, while hallucinations are effectively suppressed.

## 4 EXPERIMENTS

### 4.1 SETUP

**Baselines.** We first apply PruneHal on various decoding methods, including greedy decoding, nucleus sampling, and beam search. Besides, following previous work(Wang et al., 2024c), we also integrate PruneHal with various decoding strategies for mitigating hallucination, including DoLa (Chuang et al., 2023), VCD (Leng et al., 2024), OPERA (Huang et al., 2024) and DeCo (Wang et al., 2024c). For all aforementioned decoding strategies, we use the default hyperparameters from the source code for fair comparison.

**Model selection.** Building upon the experiments of previous works (Huang et al., 2024; Wang et al., 2024c) and to evaluate the generalizability of our method, we conduct experiments on the following models: LLaVA-v1.5-7B, LLaVA-v1.5-13B (Liu et al., 2023b), InstructBLIP-7B (Dai et al., 2023), and Qwen-VL-7B (Bai et al., 2023). These models include both MLP and QFormer (Li et al., 2023a) connectors, as well as variants of different sizes, which allows for a comprehensive demonstration of our method's effectiveness and robustness.

**Settings.** We conduct all our experiments on a single NVIDIA H20 GPU. Due to the diverse characteristics of different MLLMs (including significant variations in the number of visual tokens and the proportion of attention they receive), we specify tailored parameters for each model. For LLaVA-v1.5-7B and LLaVA-v1.5-13B, we set $r = 0.4$ and $t = 3$; for InstructBLIP-7B, we set $r = 0.7$ and $t = 2$; for Qwen-VL-7B, we set $r = 0.9$ and $t = 4$.

**Benchmarks and Metrics.** Since our framework only affects the decoding phase during the inference of MLLMs and the prefilling phase remains unchanged, we cannot use metrics that require the model to answer only yes or no, such as POPE (Li et al., 2023b), since models' responses will keep unchanged. We use CHAIR (Rohrbach et al., 2018), AMBER (Wang et al., 2023), and GPT-4V-assisted evaluation as hallucination metrics to assess our framework, with detailed settings provided in Appendix. A.

### 4.2 MAIN RESULTS

As shown in Tab. 1, across all models and decoding methods, our proposed PruneHal improves the performance of the models on both the CHAIR$_S$ and CHAIR$_I$ metrics. When integrating with existing decoding strategies for hallucination mitigation, our method further improves their performances. For example, on LLaVA-v1.5-7B, the CHAIR$_S$ metric shows that our PruneHal reduces the proportion of hallucinated sentences by 21.1%, 22.9%, and 25.0% under greedy, nucleus, and beam search, respectively. When combined with Deco, which is the current state-of-the-art approach, it achieves an additional improvement of 23.9%, 16.7%, and 18.0% over Deco alone under greedy, nucleus, and beam search, respectively, providing strong evidence for the effectiveness of our method.

On AMBER dataset, as shown in Tab. 2, after applying our method, on the three hallucination-related metrics—*CHAIR*, *Hal*, and *Cog*—the models' performance show a significant improvement, demonstrating that our method enhances the accuracy and reliability of the model's outputs.

As for GPT-4V assisted evaluation, as shown in Tab. 3, our method significantly improves the correctness of the model outputs, while maintaining the detailedness of the outputs. On LLaVA-v1.5-7B, our PruneHal improves the correctness score from 6.04 to 6.98, demonstrating a significant enhancement in the truthfulness of the model's output. The improvement in correctness indicates

a successful reduction in MLLMs' hallucinations, while the preservation of detailedness highlights that our PruneHal does not compromise the diversity of the model's responses.

## 4.3 MODEL ANALYSIS

**Analysis on hyperparameters.** We conducted experiments on different hyperparameter settings and the results are shown in Appendix. B, which shows robustness of our framework.

Table 1: **CHAIR** object hallucination evaluation results. Lower scores mean fewer hallucinations.

| Decoding | Method | LLaVA-v1.5-7B | | InstructBLIP-7B | | Qwen-VL-7B | | LLaVA-v1.5-13B | |
|---|---|---|---|---|---|---|---|---|---|
| | | $CHAIR_S$ ↓ | $CHAIR_I$ ↓ | $CHAIR_S$ ↓ | $CHAIR_I$ ↓ | $CHAIR_S$ ↓ | $CHAIR_I$ ↓ | $CHAIR_S$ ↓ | $CHAIR_I$ ↓ |
| Greedy | Vanilla | 44.6 | 12.5 | 60.0 | 24.2 | 53.6 | 14.7 | 44.2 | 12.1 |
| | **PruneHal** | **35.2** ↓9.4 | **10.0** ↓2.5 | **52.8** ↓7.2 | **23.3** ↓0.9 | **52.4** ↓1.2 | **13.5** ↓1.2 | **34.6** ↓9.6 | **9.4** ↓2.7 |
| | DoLa[ICLR'24] | 43.6 | 12.7 | 48.6 | 16.0 | 58.4 | 16.4 | 46.4 | 15.3 |
| | **DoLa + PruneHal** | **36.2** ↓7.4 | **10.5** ↓2.2 | **44.3** ↓4.3 | **14.7** ↓1.3 | **51.4** ↓7.0 | **13.8** ↓2.6 | **39.0** ↓7.4 | **13.6** ↓1.7 |
| | DeCo[ICLR'25] | 36.8 | 10.3 | 41.0 | 14.4 | 48.0 | 12.8 | 41.0 | 11.1 |
| | **DeCo + PruneHal** | **28.0** ↓8.8 | **8.7** ↓1.6 | **38.8** ↓2.2 | **13.5** ↓0.9 | **40.8** ↓7.2 | **11.4** ↓1.4 | **32.6** ↓8.4 | **9.7** ↓1.4 |
| Nucleus | Vanilla | 53.2 | 15.3 | 57.8 | 25.7 | 54.4 | 14.8 | 53.8 | 14.8 |
| | **PruneHal** | **41.0** ↓12.2 | **12.4** ↓2.9 | **49.4** ↓8.4 | **25.2** ↓0.5 | **52.8** ↓1.6 | **14.1** ↓0.7 | **45.2** ↓8.6 | **13.2** ↓1.6 |
| | VCD[CVPR'24] | 54.6 | 16.3 | 58.0 | 16.9 | 54.0 | 12.9 | 53.6 | 14.6 |
| | **VCD + PruneHal** | **41.2** ↓13.4 | **12.4** ↓3.9 | **54.0** ↓4.0 | **14.8** ↓2.1 | **51.2** ↓2.8 | **11.7** ↓1.2 | **42.4** ↓11.2 | **11.6** ↓3.0 |
| | DeCo | 40.8 | 11.0 | 45.2 | 13.4 | 48.6 | 13.0 | 41.8 | 11.9 |
| | **DeCo + PruneHal** | **34.0** ↓6.8 | **9.8** ↓1.2 | **40.6** ↓4.6 | **11.7** ↓1.7 | **46.6** ↓2.0 | **12.4** ↓0.6 | **33.8** ↓8.0 | **10.2** ↓1.7 |
| Beam Search | Vanilla | 48.8 | 13.9 | 54.0 | 15.4 | 42.2 | 10.5 | 47.8 | 13.7 |
| | **PruneHal** | **36.6** ↓12.2 | **10.4** ↓3.5 | **49.4** ↓4.6 | **14.1** ↓1.3 | **38.4** ↓3.8 | **9.8** ↓0.7 | **38.0** ↓9.8 | **10.7** ↓3.0 |
| | OPERA[CVPR'24] | 45.2 | 13.2 | 44.6 | 14.2 | 34.6 | 9.2 | 35.6 | 11.2 |
| | **OPERA + PruneHal** | **35.2** ↓10.0 | **10.2** ↓3.0 | **39.8** ↓4.8 | **12.8** ↓1.4 | **31.0** ↓3.6 | **8.6** ↓0.6 | **29.6** ↓6.0 | **9.5** ↓1.7 |
| | DeCo | 35.6 | 9.5 | 44.5 | 12.8 | 35.8 | 9.7 | 38.0 | 10.7 |
| | **DeCo + PruneHal** | **29.2** ↓6.4 | **8.0** ↓1.5 | **41.8** ↓2.7 | **11.5** ↓1.3 | **33.2** ↓2.6 | **9.3** ↓0.4 | **31.2** ↓6.8 | **9.0** ↓1.7 |

Table 2: Results on **AMBER** image captioning dataset. Lower CHAIR, Hal, and Cog values indicate better truthfulness. G., N., and B. represents Greedy, Nucleus and Beam Search, respectively.

| Dec. | Method | LLaVA-v1.5-7B | | | InstructBLIP-7B | | | Qwen-VL-7B | | | LLaVA-v1.5-13B | | |
|---|---|---|---|---|---|---|---|---|---|---|---|---|---|
| | | CHAIR↓ | Hal↓ | Cog↓ | CHAIR↓ | Hal↓ | Cog↓ | CHAIR↓ | Hal↓ | Cog↓ | CHAIR↓ | Hal↓ | Cog↓ |
| G. | Vanilla | 6.5 | 30.6 | 3.3 | 20.6 | 60.5 | 8.0 | 11.0 | 52.5 | 6.1 | 6.2 | 29.3 | 3.0 |
| | **PruneHal** | **6.2** ↓0.3 | **26.7** ↓3.9 | **2.9** ↓0.4 | **20.0** ↓0.6 | **58.6** ↓1.9 | **7.2** ↓0.8 | **10.5** ↓0.5 | **50.8** ↓1.7 | **5.6** ↓0.5 | **6.0** ↓0.2 | **28.0** ↓1.3 | **2.9** ↓0.1 |
| N. | Vanilla | 9.4 | 40.3 | 4.2 | 23.8 | 67.9 | 8.4 | 12.0 | 53.2 | 5.6 | 9.0 | 38.5 | 3.5 |
| | **PruneHal** | **8.8** ↓0.6 | **37.1** ↓3.2 | **4.0** ↓0.2 | **22.6** ↓1.2 | **64.7** ↓3.2 | **7.3** ↓1.1 | **11.5** ↓0.5 | **51.2** ↓2.0 | **5.1** ↓0.5 | **8.6** ↓0.4 | **37.2** ↓1.3 | **3.3** ↓0.2 |
| B. | Vanilla | 7.8 | 30.5 | 3.5 | 11.0 | 46.0 | 5.7 | 6.7 | 32.5 | 3.1 | 7.3 | 30.0 | 3.5 |
| | **PruneHal** | **7.4** ↓0.4 | **28.4** ↓2.1 | **3.2** ↓0.3 | **10.0** ↓1.0 | **45.2** ↓0.8 | **5.2** ↓0.5 | **6.5** ↓0.2 | **31.9** ↓0.6 | **3.0** ↓0.1 | **7.0** ↓0.3 | **26.7** ↓3.3 | **3.2** ↓0.3 |

**Analysis on adaptive module design.** Adaptive module balances crucial visual information loss and attention to key tokens. Excessive pruning harms detailedness and diversity of outputs, while insufficient pruning weakens hallucination mitigation. Experiments against simple KV cache pruning across different ratios confirm this trade-off. We evaluate hallucinations with CHAIR and diversity with the GPT-4V Detailedness metric and MM-Vet (Yu et al., 2023).

Table 3: GPT-4V assisted evaluation results. Two aspects are verified: correctness (C) and detailedness (D). Higher correctness indicates less hallucination. LLaVA-7B, BLIP-7B, Qwen-7B and LLaVA-13B indicates LLaVA-v1.5-7B, InstructBLIP-7B, Qwen-VL-7B and LLaVA-v1.5-13B, respectively.

| Model | LLaVA-7B | | BLIP-7B | | Qwen-7B | | LLaVA-13B | |
|---|---|---|---|---|---|---|---|---|
| | $C$ ↑ | $D$ ↑ | $C$ ↑ | $D$ ↑ | $C$ ↑ | $D$ ↑ | $C$ ↑ | $D$ ↑ |
| Greedy | 6.04 | 6.76 | 5.88 | 5.94 | 6.82 | 6.68 | 6.58 | 7.02 |
| **PruneHal** | **6.98** | 6.73 | **6.04** | 5.96 | **7.10** | 6.72 | **6.84** | 7.04 |

As shown in Tab. 4, conservative pruning results in high CHAIR scores (indicating persistent hallucinations), while aggressive pruning reduces hallucinations but degrades GPT4V-D and MM-Vet. In contrast, PruneHal strikes a balanced trade-off, achieving improvements on both metrics.

Table 4: Ablation on adaptive module design. For Conservative and Aggressive, pruning is applied once at the second decoding step. Conservative keeps ratio $r$, while Aggressive keeps ratio $r^t$, which equals the maximum pruning extent for each model.

| Method | LLaVA-v1.5-7B | | | | InstructBLIP-7B | | | |
|---|---|---|---|---|---|---|---|---|
| | CHAIR$_S$↓ | CHAIR$_I$↓ | GPT4V-D↑ | MM-Vet↑ | CHAIR$_S$↓ | CHAIR$_I$↓ | GPT4V-D↑ | MM-Vet↑ |
| Vanilla | 44.6 | 12.5 | 6.76 | 28.3 | 60.0 | 24.2 | 5.94 | 24.1 |
| Conservative | 40.8 | 11.9 | 6.72 | 29.5 | 56.8 | 23.9 | 5.97 | 23.6 |
| Aggressive | 37.2 | 11.1 | 6.36 | 26.2 | 54.8 | 22.9 | 5.74 | 22.5 |
| PruneHal | 35.2 | 10.0 | 6.73 | 29.4 | 52.8 | 23.3 | 5.96 | 23.7 |

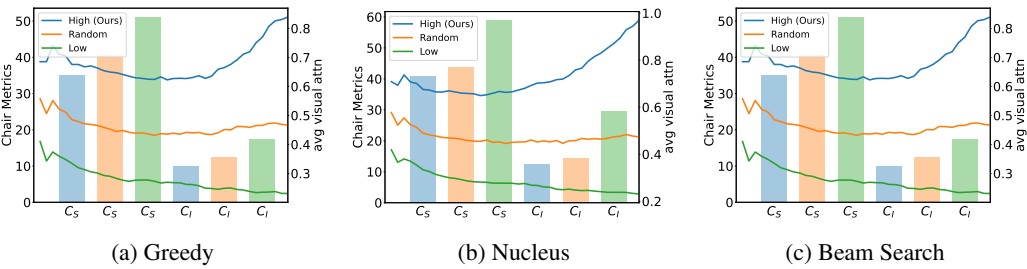

| (a) Greedy | (b) Nucleus | (c) Beam Search |
|---|---|---|

Figure 5: Average visual token attentions and hallucinations in LLaVA-v1.5-7B. Blue, orange, and green denote PruneHal (top-attended tokens), random tokens, and least-attended tokens, respectively. Lines show average attention scores over 500 samples across decoding steps; bars report CHAIR$_S$ and CHAIR$_I$, where lower values indicate fewer hallucinations.

**Analysis on visual attention scores.** In PruneHal, we retain the most attended visual tokens. To study the link between visual attention and hallucinations, we compare against retaining random tokens or the least attended ones, finding that lower attention leads to more hallucinations. As shown in Fig. 5, pruning by retaining visual tokens with the lowest attention scores significantly increases the proportion of hallucinated vocabulary (i.e. CHAIR$_I$) in the generated content. Under greedy, nucleus, and beam search, the increases are 58.7% (10.9 → 17.3), 138.71% (12.4 → 29.6), and 53.85% (10.4 → 16.0), respectively. This further underscores the importance of retaining highly-attended key visual tokens.

**Latency analysis.** Our method adds negligible overhead compared to baselines. On 100 random MSCOCO-2014 validation samples, we measured average forward-pass latency of all output tokens with default settings. Under beam search, our approach even accelerates inference, since pruning reduces the heavy computational cost associated with forward propagation. In contrast, Deco and DoLa introduce extra overhead from intermediate-layer processing, while VCD and OPERA requires multiple forward passes, yielding a marked decrease in inference speed. Our method only involves lightweight tensor operations and even reduces FLOPs, making it the most efficient.

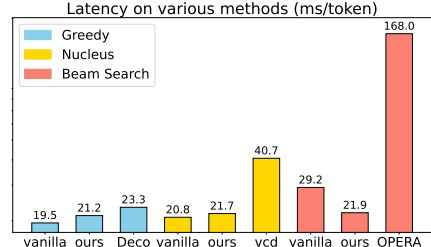

Figure 6: Per-token latency for various methods on LLaVA-v1.5-7B.

## 5 CONCLUSION

In this paper, we reveal the strong correlation between low attention to visual tokens in MLLMs and object hallucination, and propose to leverage KV cache pruning to reduce such issues. We introduce PruneHal, a dynamic KV cache pruning framework that effectively mitigates hallucinations across multiple mainstream models, while compatible with specifically designed decoding strategies for hallucination mitigation to further enhance their performance. Moreover, our method is training-free and introduces virtually no computational overhead, enabling a seamless and cost-free reduction of hallucinations in MLLMs.

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

# A EXPERIMENTAL DETAILS

## A.1 CHAIR

Caption Hallucination Assessment with Image Relevance (CHAIR) is the most widely used hallucination identification metric. CHAIR calculates what proportion of words generated are actually in the image according to the ground truth sentences and object segmentations, and evaluates object hallucination both at the instance level ($\text{CHAIR}_I$) and sentence level ($\text{CHAIR}_S$) as shown in Eq. 7. we follow the same experimental settings as OPERA (Huang et al., 2024) and Deco (Wang et al., 2024c), using the consistent 500 images from MSCOCO 2014 validation dataset, with prompt `"Please describe this image in detail."`.

$$\text{CHAIR}_I = \frac{|\{\text{hallucinated objects}\}|}{|\{\text{all mentioned objects}\}|} \quad , \quad \text{CHAIR}_S = \frac{|\{\text{captions with hallucinated objects}\}|}{|\{\text{all captions}\}|} \tag{7}$$

## A.2 AMBER

AMBER is an LLM-free, multi-dimensional benchmark designed to evaluate existence, attribute, and relation hallucinations. It includes four metrics: *CHAIR*, *Cover*, *Hal*, and *Cog*. Among these four metrics, *Cover* evaluates the comprehensiveness of model outputs, while the others assess object hallucinations from different perspectives. In our experiments, we select *CHAIR*, *Hal*, and *Cog*, the three metrics related to hallucination evaluation to assess MLLMs' hallucinations.

After generating responses, AMBER first extracts nouns from the sentences using language toolkits (e.g., NLTK). The proportion of extracted nouns that do not appear in the annotated words is calculated as the *CHAIR* metric. *Hal* metric measures the proportion of responses containing hallucinations, while *Cog* metric evaluates whether these hallucinations resemble those found in human cognition. By leveraging a set of hallucinatory target objects, the likelihood of MLLMs generating these objects is computed.

The AMBER dataset contains 1,004 images across diverse object categories, with 14 major categories, such as Nature, Architecture, and Street View. The distribution of each category is fairly

balanced across categories without a long-tail phenomenon. The prompt used for the evaluation is:
`"Describe this image."`.

## A.3  GPT-4V ASSISTED EVALUATION

Following prior work (Huang et al., 2024), we conduct an open-ended evaluation with GPT-4V on 500 randomly sampled COCO images. GPT-4V compares the outputs of two assistants with respect to Correctness (C) (i.e., truthfulness) and Detailedness (D) (i.e., richness). The two answers from vanilla models and the models paralleled with our PruneHal framework are offered to GPT-4V at the same time for fair comparison, and it is required to give a judgement ranging from 1 to 10 points respectively for each metric. The prompts are provided in Tab. 5.

---

**GPT-4V prompt**

---

You are required to score the performance of two AI assistants in describing a given image. You should pay extra attention to the hallucination, which refers to the part of descriptions that are inconsistent with the image content, such as claiming the existence of something not present in the image or describing incorrectly in terms of the counts, positions, or colors of objects in the image. Please rate the responses of the assistants on a scale of 1 to 10, where a higher score indicates better performance, according to the following criteria:

1: Accuracy: whether the response is accurate with respect to the image content. Responses with fewer hallucinations should be given higher scores.

2: Detailedness: whether the response is rich in necessary details. Note that hallucinated descriptions should not count as necessary details.

Please output the scores for each criterion, containing only two values indicating the scores for Assistant 1 and 2, respectively. The two scores are separated by a space. Following the scores, please provide an explanation of your evaluation, avoiding any potential bias and ensuring that the order in which the responses were presented does not affect your judgment.

[Assistant 1]
{}
[End of Assistant 1]
[Assistant 2]
{}
[End of Assistant 2]
Output format:
Accuracy: <Scores of the two answers >
Reason:
Detailedness: <Scores of the two answers >
Reason:

---

Table 5: Prompt for GPT-4V assisted evaluation.

## B  ABLATION STUDIES ON HYPERPARAMETERS

We conduct hyperparameter ablation experiments on LLaVA-v1.5-7B and InstructBLIP-7B, using CHAIR metrics on using the consistent 500 images from MSCOCO 2014 validation dataset. The results show that both parameters in PruneHal ($r$ and $t$) exhibit strong robustness and can consistently reduce hallucinations within a reasonable range. The results are shown in Tab. 6.

Table 6: Ablation study on hyperparameters. Our selected hyperparameters and the best results are highlighted in bold.

| LLaVA-v1.5-7B | Greedy | | Nucleus | | Beam Search | |
|---|---|---|---|---|---|---|
| | $C_S\downarrow$ | $C_I\downarrow$ | $C_S\downarrow$ | $C_I\downarrow$ | $C_S\downarrow$ | $C_I\downarrow$ |
| Vanilla | 44.6 | 12.5 | 53.2 | 15.3 | 48.8 | 13.9 |
| **r=0.4, t=3** | 35.2 | 10.0 | **41.0** | 12.4 | **36.6** | **10.4** |
| r=0.4, t=2 | 38.4 | 11.1 | 41.2 | **12.3** | 38.2 | 10.7 |
| r=0.5, t=3 | 38.2 | **10.5** | 41.6 | 12.8 | 38.6 | 10.7 |
| r=0.5, t=2 | 41.8 | 11.6 | 47.6 | 13.7 | 41.2 | 12.3 |
| r=0.3, t=3 | **37.4** | 10.8 | 43.2 | 14.0 | 36.2 | 10.7 |
| r=0.3, t=2 | 36.7 | 10.9 | 43.2 | 10.8 | 36.6 | 10.8 |

| InstructBLIP-7B | Greedy | | Nucleus | | Beam Search | |
|---|---|---|---|---|---|---|
| | $C_S\downarrow$ | $C_I\downarrow$ | $C_S\downarrow$ | $C_I\downarrow$ | $C_S\downarrow$ | $C_I\downarrow$ |
| Vanilla | 60.0 | 24.2 | 57.8 | 25.7 | 54.0 | 15.4 |
| **r=0.7, t=2** | **52.8** | 23.3 | **49.4** | 25.2 | **49.4** | **14.1** |
| r=0.7, t=3 | 52.9 | 23.4 | 50.3 | 25.1 | 50.1 | 14.8 |
| r=0.8, t=2 | 57.0 | **22.9** | 50.2 | 24.4 | 53.2 | 14.7 |
| r=0.8, t=3 | 56.2 | 23.7 | 54.4 | 26.0 | 51.4 | 14.7 |
| r=0.6, t=2 | 57.4 | 23.8 | 51.2 | **23.5** | 53.2 | 15.1 |
| r=0.6, t=3 | 59.8 | 24.7 | 56.9 | 25.4 | 51.8 | 15.2 |

## C DETAILED EXPERIMENTAL SETTINGS AND MORE EXAMPLES IN SECTION 1

For experiments conducted in Fig. 1 and Fig. 2, following (Huang et al., 2024), the images are selected from the 500 images subset from MSCOCO 2014 validation dataset, and all experiments are conducted on LLaVA-v1.5-7B. The prompts and images are exactly the same as in Appendix. A.1. We first acquire the generated results and get image-caption pairs, and selected those captions containing hallucinatory outputs.

### C.1 DETAILED SETTINGS FOR QUALITATIVE EXPERIMENT

For the experiment mentioned in Fig. 1, We truncate the generated text up to (but not including) the hallucinated word, and feed it into the model together with the original instruction. In each layer's self-attention module of the language model, after computing the attention map, we double the attention values corresponding to visual tokens at the position of the last token. Once the forward pass produces an output token, we append this token to the instruction. For the following tokens, no further amplification of attention is applied, and the computation proceeds normally.

### C.2 DETAILED SETTINGS FOR QUANTITATIVE EXPERIMENT

For the experiment mentioned in Fig. 2, we select all image-caption pairs containing hallucinatory outputs (223 out of 500). For captions containing hallucinations, we track visual attention scores for the entire generation process. At each decoding step, we compute the average attention score from the generated token to all visual tokens. After collecting the data for all tokens, we calculate both the overall average across all tokens (shown as a single red scatter in Fig. 2) and the average specifically for the hallucinated tokens (shown as a single red scatter in Fig. 2).

### C.3 MORE EXAMPLES FOR QUALITATIVE EXPERIMENT

We also provide more examples for the qualitative experiment in Sec. 1 in Fig. 7.

### C.4 MORE EXAMPLES FOR EXPERIMENTS IN SEC. 3.2

In Fig. 3, we presented examples from the first layer of the language models in various MLLMs. In Fig. 8, we additionally report results from Layers 16 and 31, representing middle and deep layers in 32-layer language models, respectively. These layers exhibit the same phenomenon as Fig. 3, highlighting the generalizability of our findings.

## D CASES

In this section, we present several cases where our proposed PruneHal successfully mitigates hallucinations in MLLMs. By contrasting these successful cases with the outputs of the vanilla model, we demonstrate that PruneHal not only suppresses hallucinations but also preserves the overall descriptive quality of the generated responses.

We conduct experiments under greedy search mode for every model. The cases for LLaVA-v1.5-7B, Qwen-VL-7B, InstructBLIP-7B and LLaVA-v1.5-13B are shown in Figs. 9–12, respectively.

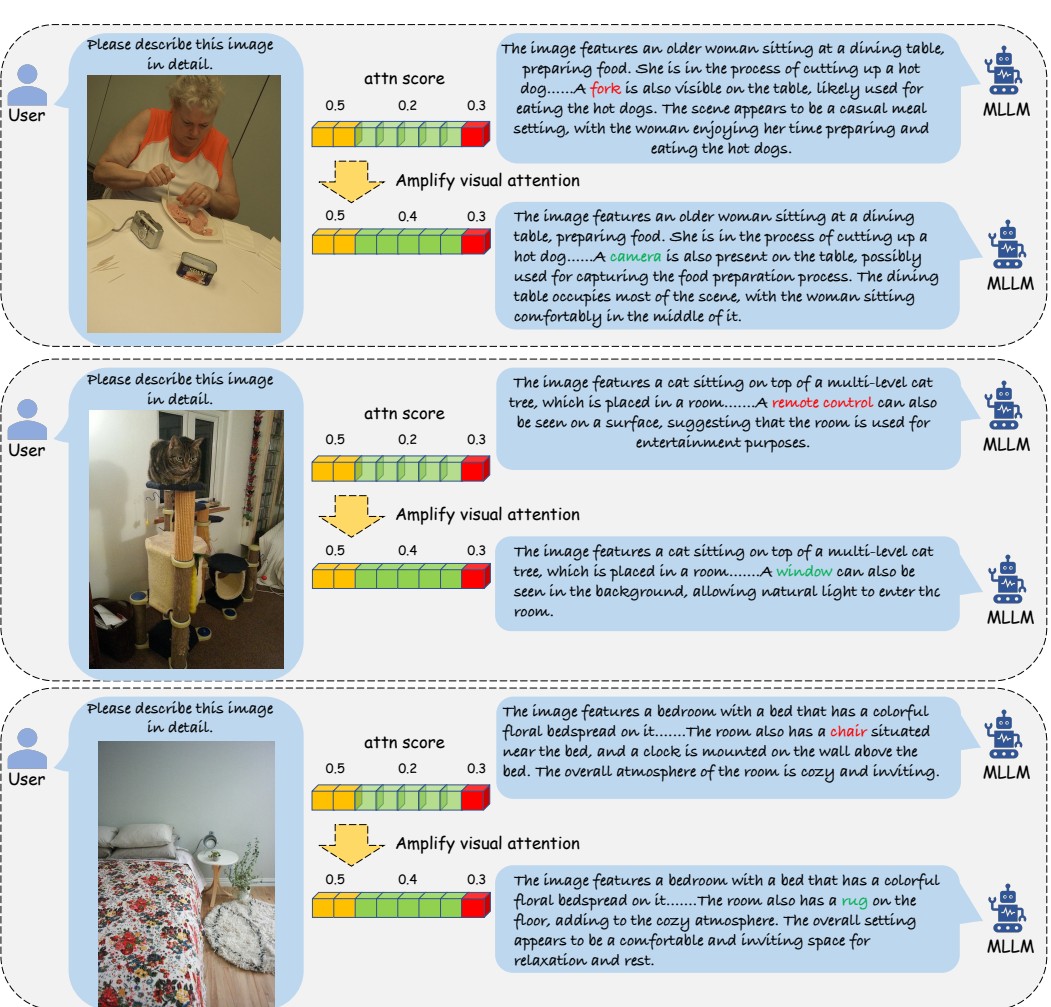

Figure 7: Additional examples for qualitative experiments in Sec. 1.

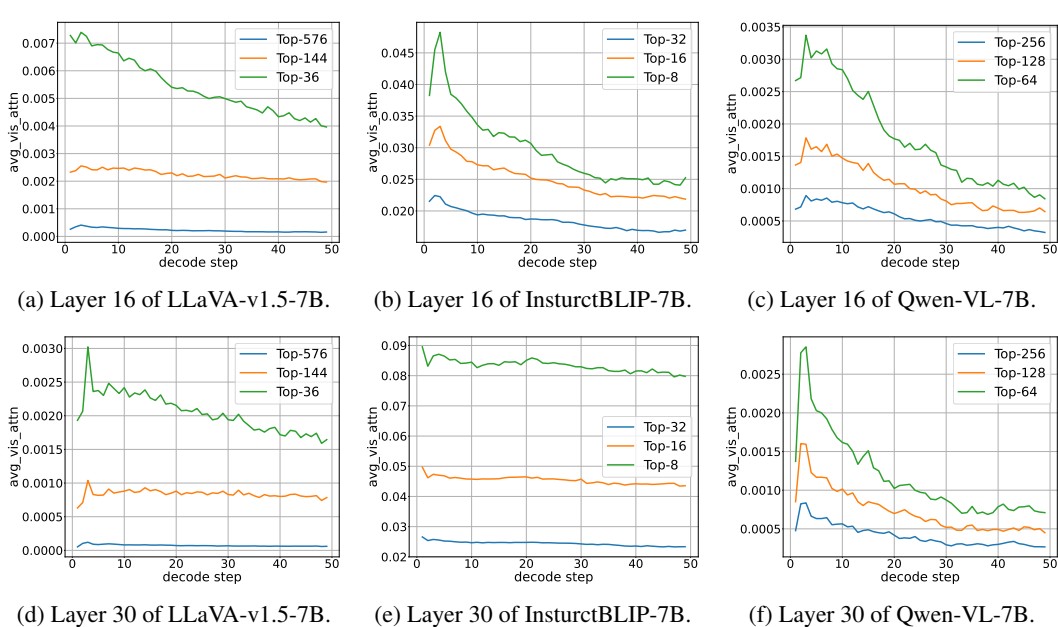

(a) Layer 16 of LLaVA-v1.5-7B.

(b) Layer 16 of InsturctBLIP-7B.

(c) Layer 16 of Qwen-VL-7B.

(d) Layer 30 of LLaVA-v1.5-7B.

(e) Layer 30 of InsturctBLIP-7B.

(f) Layer 30 of Qwen-VL-7B.

Figure 8: Examples from middle and deep layers for experiments in Sec. 3.2.

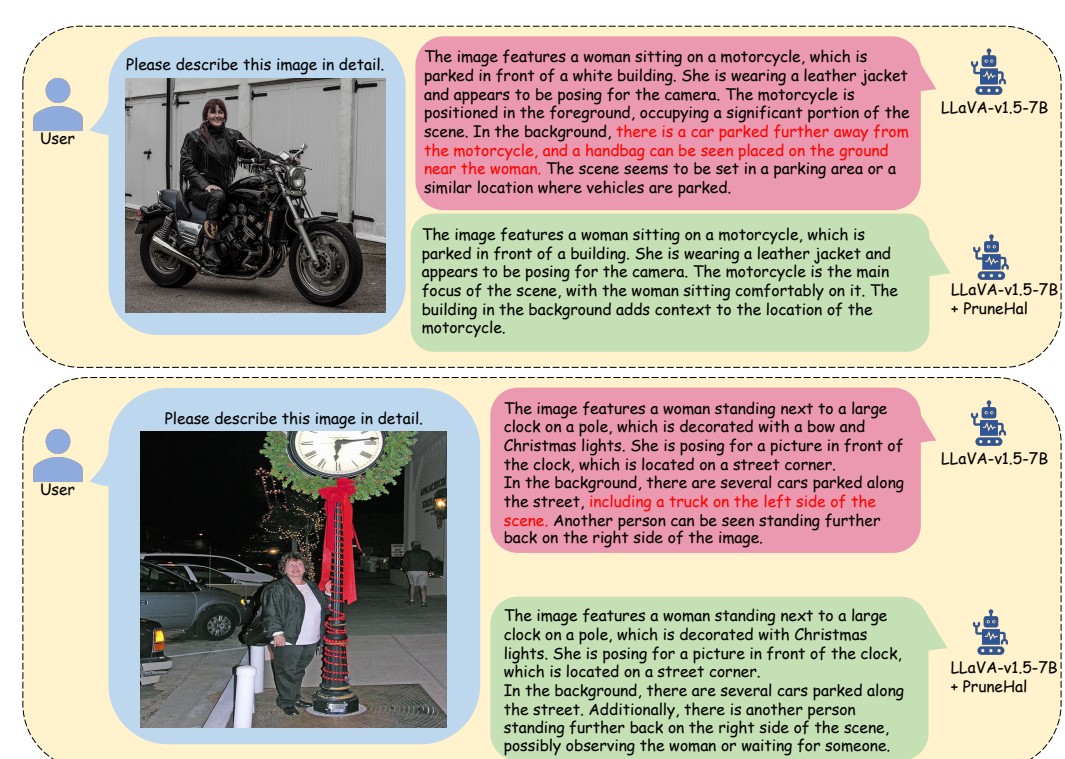

Figure 9: Cases for LLaVA-v1.5-7B.

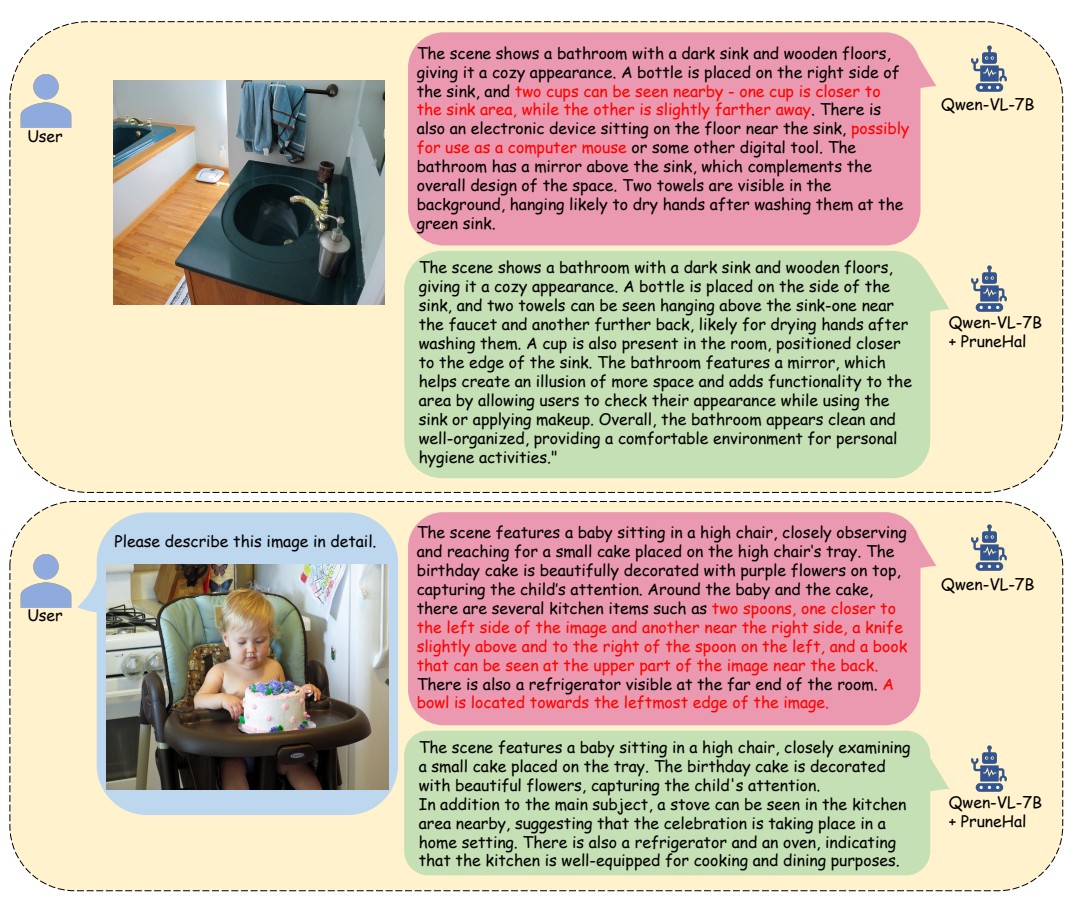

Figure 10: Cases for Qwen-VL-7B.

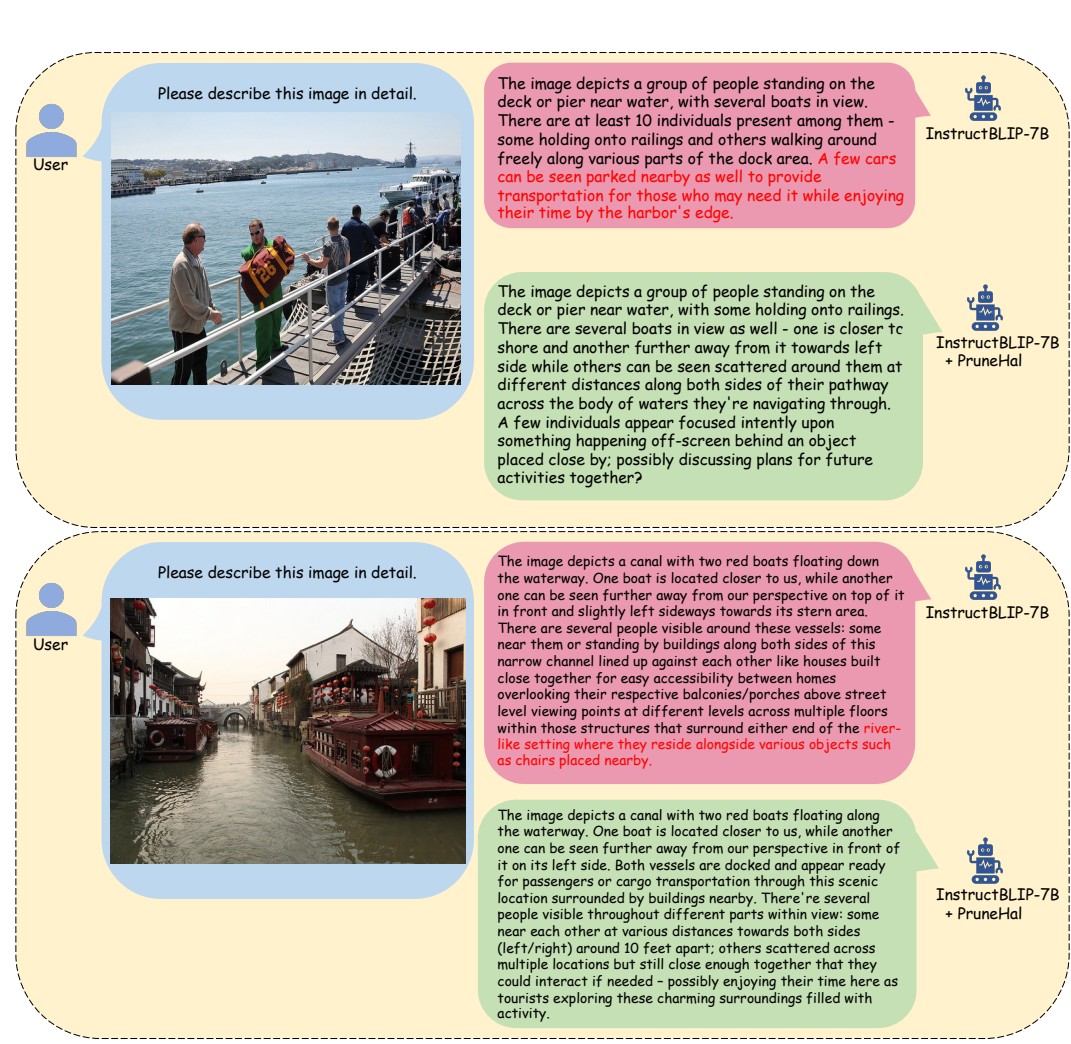

Figure 11: Cases for InstructBLIP-7B.

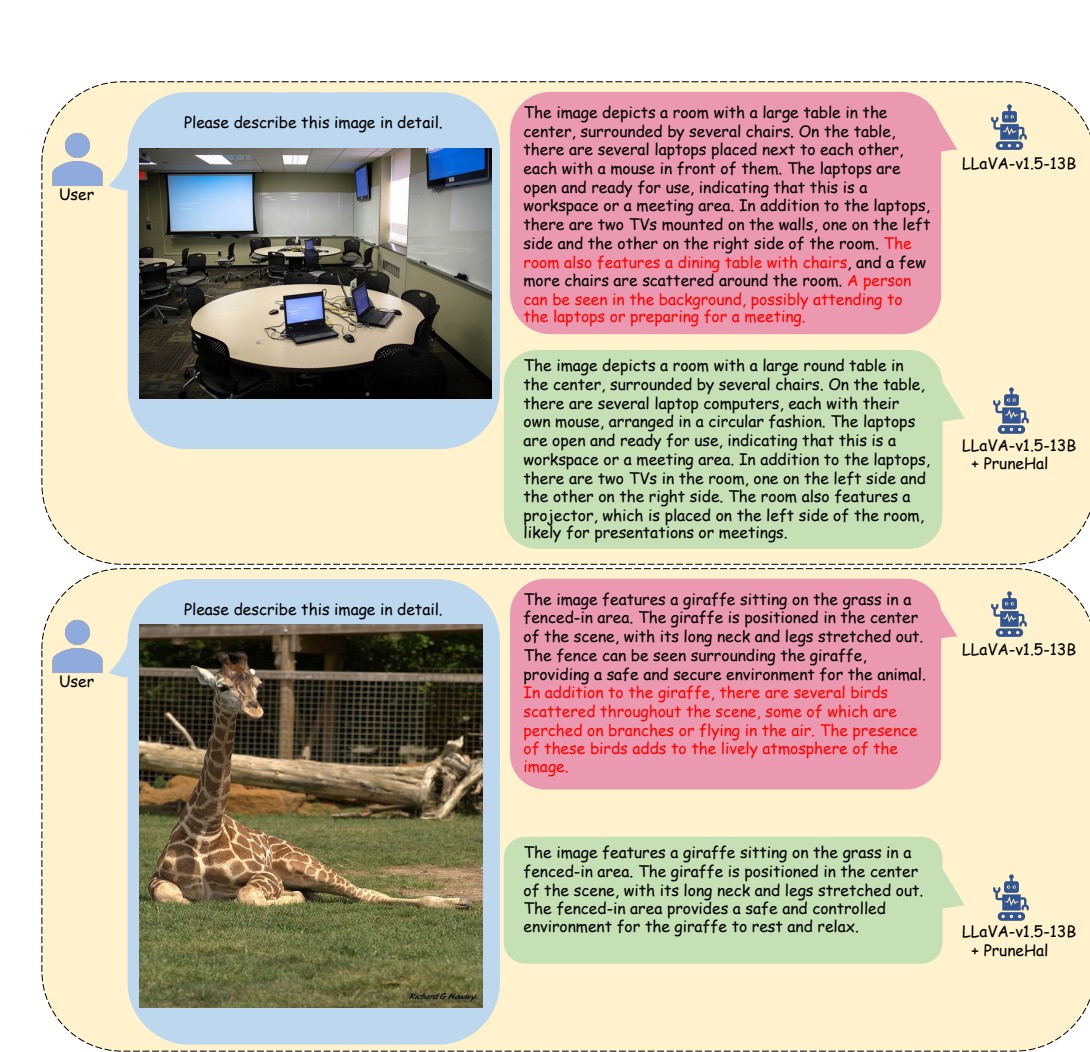

Figure 12: Cases for LLaVA-v1.5-13B.

