# OpenReview forum: "PruneHal: Reducing Hallucinations in Multi-modal Large Language Models through Adaptive KV Cache Pruning"
_ICLR.cc/2026/Conference — ICLR 2026 Conference Withdrawn Submission_

### Official Review · Reviewer_HCd2 · 2025-10-16

**Soundness:** 2
**Presentation:** 3
**Contribution:** 2
**Rating:** 2
**Confidence:** 4

**Summary:**

This paper introduces a novel approach: PruneHal. It reframes hallucination mitigation as attention concentration on informative visual evidence via adaptive, training-free KV-cache pruning, and reduces hallucinations across diverse MLLMs/decoders while maintaining descriptive detail and incurring minimal inference cost. Detailed information is as follows:

## 1) Research Gap
MLLMs still hallucinate (generate content inconsistent with the image) despite strong overall capability.
Existing fixes usually need extra data/training or specialized decoding, increasing cost/latency.
Under-explored angle: long, redundant visual token sequences; low attention to visual tokens correlates with hallucination during generation.
## 2) Motivation
Qualitative probe: boosting attention to visual tokens near an error can flip an incorrect word to a correct one.
Quantitative probe: at steps producing hallucinations, average attention to visual tokens is typically below the caption’s baseline (mean across steps).
Goal: concentrate attention on salient visual evidence during inference without retraining.
## 3) Proposed Method (PruneHal)
Training-free, plug-and-play: adaptively prunes only the visual entries in the decoder’s KV cache; text KV is unchanged.
Signal: use the most recent step’s attention over visual tokens (saliency).
Trigger (layer-vote): for each layer $\ell$, compare current average visual-token attention $A_\ell^{(i)}$ to a running historical average $\bar{A}\ell$. If a majority of layers satisfy $A_\ell^{(i)} < \sqrt{r}\bar{A}_\ell$ , pruning is triggered.
Action (fractional keep): retain the top-$r$ fraction of visual tokens (by attention from the previous step) and physically slice the visual KV to those indices; discarded tokens do not re-enter.
Safeguards: cap the number of pruning operations at $t$; use $r\in(0,1)$ (not a tiny fixed budget) to keep buffer tokens for potential late-use content.
A fixed Top-$K$ variant exists; the adaptive version better balances truthfulness vs. descriptive richness.
## 4) Experimental Results
Models/Decoding: LLaVA-v1.5-7B/13B, InstructBLIP-7B, Qwen-VL-7B; greedy, nucleus, beam; also combined with DoLa, VCD, OPERA, DeCo.
Benchmarks: CHAIR, AMBER facets, MM-Vet, plus LLM-assisted open-ended evaluation (correctness/detailedness).
Findings:
* Hallucination reduction: consistent CHAIR decreases across models/decoding strategies.
* Compatibility: complementary to other decoding methods; gains add up.
* Quality balance: correctness improves while detailedness is largely preserved.
* Efficiency: negligible overhead; under beam search, pruning can reduce per-token latency by shrinking the visual attention set.
## 5) Analysis
Token selection: keeping top-attended visual tokens reduces hallucinations more than random or low-attention selections.
Adaptive vs. fixed: aggressive one-shot pruning curbs hallucinations but can harm richness; conservative fixed pruning under-mitigates hallucinations. The adaptive trigger (layer vote + $\sqrt{r}$ threshold) with cap $t$ strikes a better trade-off.
Hyperparameter robustness: reasonable ranges of $(r, t)$ yield consistent gains with light model-specific tuning.
Risk & mitigation: pruning is irreversible; premature pruning could drop late-use evidence. Fractional keep, the layer-voted trigger, and prune cap mitigate this in practice.

**Strengths:**

## 1. Training-free, plug-and-play, and compatible with other methods
   The method requires no retraining, slots into standard decoding, and is explicitly positioned as compatible with hallucination-aware decoders. The conclusion also emphasizes “virtually no computational overhead.”


## 2. Adaptive, principled trigger that tracks attention drift
   The design of this paper reacts only when a majority of layers’ visual-token attention falls below a threshold (layer-vote with a ($\sqrt{r}$) criterion), motivated by the observation that visual attention can continuously decline during long generations. The algorithm specifies the vote, threshold, and bounded number of prunes.


## 3. Consistent hallucination reduction across models and decoding strategies
   On AMBER/CHAIR, PruneHal improves LLaVA-v1.5-7B/13B, InstructBLIP-7B, and Qwen-VL-7B over greedy, nucleus, and beam decoding; tables show the deltas for CHAIR, Hal, and Cog.


## 4. Quality balance: improves correctness while preserving detail
   The paper reports GPT-4V-assisted Correctness gains while tracking Detailedness, and frames the adaptive module as balancing hallucination mitigation with output diversity/detail. (Table and accompanying discussion.)


## 5. Efficiency: negligible overhead; can even speed up beam search
   The latency analysis shows near-zero overhead in general and faster per-token decoding under beam search because the pruned cache reduces compute, whereas alternatives like DoLa/DeCo/VCD/OPERA add extra passes or intermediate-layer work.

**Weaknesses:**

## 1. **Correlation, not causation, in the empirical motivation**
   The key quantitative probe links lower visual attention to hallucination using scatter plots and description, with details deferred to the appendix; This experiment is not considered as a full causal study, as 1) This experiment only shows the performance of llava-v1.5-7b. 2) Since visual-token attention tends to decay over later decoding steps, if hallucinations also occur later, lower attention could partially reflect step index rather than a direct cause.

## 2. **Evaluation breadth is narrow for a strong general claim**
   The main metrics center on CHAIR/AMBER for captioning and a GPT-4V judge; this focuses on object hallucination and open-ended descriptions, but leaves out other widely-used task families (e.g., VQA variants) within the main text/tables.
   The authors explicitly note **POPE** is *not used* because yes/no outputs don’t change under their setup—so an entire line of existence-probing evaluation is omitted.

## 3. **Heuristic trigger design with limited theoretical grounding**
   The adaptive trigger uses a layer vote and a $\sqrt{r}$ threshold. While intuitive, there’s no theoretical justification for $\sqrt{r}$ beyond heuristic motivation, and the analysis focuses on empirical ablations rather than principled derivation.

## 4. **Hyperparameters tuned per model**
   The paper sets different $(r, t)$ for each MLLM (e.g., LLaVA-7B/13B: $r{=}0.4,t{=}3$; InstructBLIP-7B: $r{=}0.7,t{=}2$; Qwen-VL-7B: $r{=}0.9,t{=}4$). This brings a concern that as a plug-and-play approach, when applied to a new model, does it need to conduct several experiments to allocate the best parameters before using?
   Additionally, their ablation evidences robustness on the same 500-image slice, but still within the same dataset and models.

## 5. **Sample sizes for analyses are modest**
   Several analyses use 500 images (motivation/GPT-4V) or 100 images (Top-K visualization curves), which are useful but limited for claims of breadth.

## 6. **Irreversibility and potential late-use information loss**
   The method physically discards visual KV rows, and the paper itself notes that visual-token attention diminishes as decoding proceeds, motivating a dynamic mechanism. Still, once dropped, tokens can’t return; there is no recovery strategy beyond conservative $r$ and a cap $t$.

**Questions:**

# Questions and Concerns

## A. Empirical motivation: attention vs. hallucination (Layers 1, 16, 30)

1. **Position confound (step index).** Visual-token attention typically decays over later decoding steps. If hallucinations also tend to occur later, lower attention at those steps may partially reflect position rather than a causal link. Therefore, the motivation experiment is not considered as a full causal study. Additional experiments and explanations is preferred.

2. **Model generality.** The analysis is shown only for llava-v1.5-7b. The authors are suggested to replicate the same measurement on additional MLLMs to demonstrate robustness across architectures.

3. **Caption selection bias.** Plots include only captions that already contain hallucinations (223/500). The authors may also report the distributions for non-hallucinating captions, and provide summary statistics/effect sizes and (ideally) significance tests comparing the two groups.


## B. Method: applicability and length sensitivity

1. **Trigger dependence on output length.** PruneHal’s pruning is event-triggered; very short generations may never trigger it. The authors may clarify the operating range: how often does pruning trigger vs. caption length, and report the distribution of prune counts (and time-to-first-prune) across lengths. And, if possible, a length-sensitivity plot (quality vs. output length) to show behavior on short, medium, and long captions is preferred.


## C. Evaluation scope

1. **Limited benchmarks.** Current results emphasize CHAIR/AMBER (captioning) and one GPT-judge setup. The authors are suggedted to add broader evaluations (e.g., LLaVA-Bench (In-the-Wild) and/or other open-ended sets).


## D. Baseline discrepancy (MM-Vet)

1. **Score gap for LLaVA-v1.5-7B.** The paper reports **28.3** on MM-Vet, while the LLaVA repository/paper commonly reports **31.1**. Please reconcile the difference by specifying exact evaluation settings (dataset/version, decoding parameters, resolution, conversation template), and judge model/version. If the authors follow settings that differ from the “official” configuration, consider reporting both numbers (yours and a reproduced one under their settings).


*Thanks for considering these points. I would like raise my score once my concerns has been addressed.*

**Details Of Ethics Concerns:**

No specific ethic concern.

---

### Official Review · Reviewer_wbKx · 2025-10-31

**Soundness:** 3
**Presentation:** 3
**Contribution:** 2
**Rating:** 4
**Confidence:** 4

**Summary:**

This paper introduces PruneHal, a framework for mitigating hallucinations in MLLMs.

The core idea is that hallucinations in MLLMs arise from low attention to visual tokens and redundancy in the visual input.

PruneHal applies an adaptive key-value (KV) cache pruning strategy during decoding: it uses a cross-layer attention voting mechanism to identify visual tokens that receive low attention and prunes them from the cache, thereby forcing the model to attend better to remaining visual tokens.

Several open‐source MLLMs (e.g., LLaVA-1.5, Qwen-VL) verified and reached ~25 % relative reduction in hallucination metrics.

**Strengths:**

The method is practical: no retraining, no architecture change, plug-in compatibility with existing MLLMs.

Good empirical coverage across multiple models

The topic is significant: hallucination in MLLMs is a major deployment hurdle.

**Weaknesses:**

Novelty is limited: prior works have identified the same underlying problem (neglect of visual tokens) and proposed decoding-time mitigations, such as OPERA

Narrow experimental scope: the experiments rely mainly on the CHAIR-based benchmark. Other hallucination types (e.g., relation/attribute hallucination, VQA, multi-image or video inputs, it is easy to name many benchmarks such as pope/ hallusionbench/MMhal-bench, etc) are not explored. This limits claims of generality.

Interpretability/analysis depth: While attention visualisation is provided, the paper could deepen the analysis to explain why pruning certain visual tokens can generalise to all domains.

Hyperparameter sensitivity: The pruning threshold and voting mechanism are somewhat heuristic; more systematic sensitivity analysis would strengthen confidence in robustness.

**Questions:**

Please see weaknes part

---

### Official Review · Reviewer_QcpQ · 2025-10-31

**Soundness:** 3
**Presentation:** 3
**Contribution:** 3
**Rating:** 6
**Confidence:** 4

**Summary:**

This paper identifies a factor that may be closely associated with the hallucination phenomenon in multi-modal large language models (MLLMs), specifically that redundant visual tokens disperse the model’s attention, potentially leading to hallucinations.  Based on this insight, the authors propose PruneHal, a training-free approach that only leverages adaptive KV cache pruning to enhance the model’s focus on critical visual information, thereby reducing hallucinations.

**Strengths:**

* This paper identifies a noteworthy phenomenon: the allocation of attention influences the generation of hallucinations.
* Building on this insight, PruneHal proposes a pruning-based approach to remove some redundant information, thereby reducing hallucinations.  It requires no additional training and operates at a relatively fast speed.
* In the experiments, the paper not only tests the effectiveness of its own scheme but also verifies that combining this scheme with other existing schemes can further improve performance.  This indicates that the proposed scheme is relatively versatile.

**Weaknesses:**

* The models tested in the paper are somewhat outdated. For instance, Qwen-VL, which is tested in the paper, has now been updated to Qwen3-VL, and LLaVA-v1.5 has also been updated to LLaVA-NeXT. It is suggested that more up-to-date models should be used to test the performance of PruneHal.
* Why does Algorithm 1 not explain why pruning should also be triggered when m = 2? Is it hypothesized that the authors intend to intervene in visual attention at an early stage to prevent redundant visual tokens from diverting the model's attention in the initial phase?
* The paper mentions that PruneHal accelerates inference, but only provides data for beam search. For different model scales (e.g., 7B vs. 13B) and decoding strategies (Greedy, Nucleus), what are the specific differences in inference speedup and computational cost reduction brought by PruneHal?

**Questions:**

* It seems that the argument that redundant attention allocation may exacerbate hallucinations is only supported by testing the effect of retaining tokens through pruning. In future work, supplementary verification could be added to determine whether "proactively amplifying the attention scores of critical tokens" can further optimize performance and improve the final results.
* As observed in Figure 2, there are a small number of hallucinatory cases where the Average Attention value is relatively high. Could this phenomenon be explained? If processed according to the method proposed in the paper, these hallucinatory cases may also be retained, and it remains unclear whether this has an impact on the final generation results.
* The paper applies different Top-K token numbers (e.g., Top-576/144/36 for LLaVA-v1.5-7B, Top-32/16/8 for InstructBLIP-7B, and Top-256/128/64 for Qwen-VL-7B) across models.  However, it does not explicitly explain the rationale behind these model-specific Top-K choices.  What is the specific design consideration or empirical basis for determining these distinct Top-K values for different MLLMs?

---

### Official Review · Reviewer_oCY1 · 2025-11-01

**Soundness:** 1
**Presentation:** 2
**Contribution:** 2
**Rating:** 4
**Confidence:** 5

**Summary:**

The paper identifies hallucinations in MLLMs as a major challenge and posits a link between these hallucinations and insufficient attention to critical visual tokens. The authors hypothesize that this problem is caused by a large number of redundant visual tokens, which disperse the model's attention.To address this, the paper proposes PruneHal, a training-free, adaptive KV cache pruning method. The method aims to remove redundant (low-attention) visual tokens, thereby forcing the model to concentrate its attention on the remaining, more informative ones. The proposed mechanism is adaptive, tracking the historical visual attention distribution and triggering a pruning step via a layer-wise voting mechanism when attention drops below a threshold . The authors claim this method adds "nearly no extra inference cost" , is model-agnostic , and effectively mitigates hallucinations on several benchmarks.

**Strengths:**

- The method is training-free  and highly efficient. As shown in Figure 6, it adds negligible overhead and can even accelerate inference during beam search by reducing the KV cache size.
- The idea of using adaptive pruning as a hallucination mitigation technique is novel. The layer-wise voting mechanism to decide when to prune is a smart approach to balance information loss and attention focusing.

**Weaknesses:**

- The paper's most critical weakness is its admission that it cannot be evaluated on benchmarks like POPE because "models' responses will keep unchanged". This implies the pruning mechanism only activates after the first token is generated. This is a fundamental design flaw. A hallucination mitigation strategy that cannot influence the first generated token is of very limited use, as it cannot correct a model that is already on a hallucinatory path from the very first token (e.g., answering "Yes" to a "is there a..." question when the object is absent).
- As a direct consequence of the first weakness, the paper's evaluation is incomplete. It is forced to rely only on long-form generation metrics (CHAIR, AMBER, GPT-4V). The complete and acknowledged absence of a standard object-presence benchmark like POPE is a major red flag that points to the method's limited applicability.

**Questions:**

- Please clarify the exact timing of the pruning. Does the PruneHal intervention influence the generation of the very first output token?
- If, as the text  implies, the method cannot change the first token, how can this be considered a robust hallucination mitigation strategy? Hallucinations often begin with the first word (e.g., "Yes," or "I see a...").
- The claim that POPE responses "will keep unchanged" is a critical admission. Does this mean the method is fundamentally incompatible with any task requiring a single, decisive token output (e.g., VQA)?
- Algorithm 1  appears to initialize the historical attention A using the attention from the first decoding step. Does this not confirm that pruning can, by definition, only begin at or after decoding step 2?

---

### Official Review · Reviewer_hbeo · 2025-11-01

**Soundness:** 2
**Presentation:** 3
**Contribution:** 2
**Rating:** 4
**Confidence:** 5

**Summary:**

This paper introduces PruneHal, a novel framework aimed at reducing hallucinations in Multimodal Large Language Models (MLLMs) by leveraging adaptive KV cache pruning. Hallucinations, which occur when the model generates content that diverges from the visual input, have been a significant challenge in MLLMs. Existing methods often rely on additional training or computationally expensive inference techniques. The authors propose a training-free solution that improves the model's focus on critical visual information by selectively pruning redundant visual tokens during the inference phase. The method shows promising results, reducing hallucinations with minimal computational overhead and without the need for additional training. PruneHal is also model-agnostic, capable of enhancing the performance of various decoding strategies designed to mitigate hallucinations.

**Strengths:**

1. PruneHal is a training-free solution, making it accessible for deployment without additional training costs.
2. PruneHal introduces minimal computational overhead compared to existing methods that require additional inference steps or training.
3. The paper includes informative visual attention plots and latency analysis (Figures 3 and 6), which effectively demonstrate how pruning enhances model attention on critical visual tokens while mitigating hallucinations.
4. The paper presents extensive experiments on a variety of MLLMs, including LLaVA, Qwen2-VL, InstructBLIP, and others, showing consistent improvements in hallucination reduction across different benchmarks (e.g., MME, MMMU, AI2Diagram).

**Weaknesses:**

1. The paper provides an overview of how adaptive pruning works, but does not fully explain the underlying mechanics that govern the pruning decisions.
2. The method is evaluated on a variety of MLLMs, but there is little discussion on how **PruneHal** would perform on SOTA models like Qwen2.5-VL, Qwen3-VL, or InternVL3.5.
3.  A more detailed analysis of failure cases or ablation studies on extreme pruning scenarios would be valuable.
4. Some related and important works are not included in the discussion or experimental comparison, like HALC[1], MemVR[2], RLAIF-V[3], MINT[4].
5. There are a few **minor clarity issues** and inconsistencies in notation. For example, some variables related to pruning (e.g., `r`, `t`, `Kvis`) are not fully defined or explained in detail. The presentation could benefit from tighter organization and clearer connections between figures and textual descriptions.

[1] HALC: Object Hallucination Reduction via Adaptive Focal-Contrast Decoding, ICML 2024.

[2] Look Twice Before You Answer: Memory-Space Visual Retracing for Hallucination Mitigation in Multimodal Large Language Models, ICML 2025.

[3] RLAIF-V: Open-Source AI Feedback Leads to Super GPT-4V Trustworthiness, CVPR 2025.

[4] Mitigating Hallucinations in Large Vision-Language Models via Token Reduction, 2025.02.

**Questions:**

1.  Can you elaborate on the specific decision process for adaptive pruning? How is the threshold for pruning dynamically adjusted during inference?
2. Have you considered edge cases where pruning may accidentally discard crucial visual tokens?
3. Have you considered evaluating PruneHal on tasks that require multimodal reasoning across longer contexts or temporal reasoning, e.g., video analysis?

---

### Note · Authors · 2025-11-12

**Comment:**

Thanks for all reviewers’ feedbacks. Your comments help us improve our work a lot.

**Withdrawal Confirmation:**

I have read and agree with the venue's withdrawal policy on behalf of myself and my co-authors.